# ATM-associated signalling triggers the unfolded protein response and cell death in response to stress

Yuka Hotokezaka[1], Ikuo Katayama[1] & Takashi Nakamura [1✉]

Endoplasmic reticulum (ER) stress can be caused by perturbations in ER function resulting from the accumulation of unfolded/misfolded proteins in the ER lumen. Accumulating unfolded proteins trigger unfolded protein responses (UPRs) through activating three transmembrane sensors on the ER: IRE1α, PERK, and ATF6. The orchestrated action of these molecules upregulates genes encoding proteins involved in the downregulation of protein synthesis and acceleration of protein secretion. Ineffectiveness of these fail-safe mechanisms may lead to apoptosis. However, the molecular mechanisms upstream of the UPR are not fully understood. Here we show participation of ataxia telangiectasia mutated (ATM) in stress-induced apoptosis. Cytoplasmic ATM serves as a platform on which protein phosphatase 2A-dependent dephosphorylation of AKT activates glycogen synthase kinase 3β, thereby downregulating nascent polypeptide-associated complex α subunit and γ-taxilin, triggering UPRs and leading to mitochondria-dependent apoptosis. These results suggest an ATM/AKT-dependent cell death pathway triggered by various forms of stress.

[1] Department of Radiology and Cancer Biology, Nagasaki University Graduate School of Biomedical Sciences 1-7-1 Sakamoto, Nagasaki 852-8588, Japan.
✉email: taku@nagasaki-u.ac.jp

Endoplasmic reticulum (ER) stress can occur as a result of excess accumulation of unfolded or misfolded proteins in the ER lumen. Crowding conditions in the ER lumen are detected by the three sensors at the ER plasma membrane: IRE1α, PERK, and ATF6. Under physiological conditions, the master ER lumenal chaperone-binding immunoglobulin protein (BiP) binds to and inactivates the ER sensors[1,2]. Excess accumulation of unfolded or misfolded proteins in the ER lumen sequesters BiP away from the lumenal domains of the ER sensors, freeing the sensors to be activated through homodimerization and autophosphorylation (IRE1α and PERK), or translocation from the ER to the Golgi apparatus followed by protease-mediated fragmentation (AFT6), thereby relaying the signals to the downstream unfolded protein response (UPR) pathways to alleviate the stress by facilitating the excretion of or inhibiting the synthesis of the excess proteins[3].

ER stress or UPR can occur in many physiological and pathological conditions, and is considered to contribute to the pathogenesis of several human diseases, including cancer, autoimmune diseases, diabetes, and ischemia/reperfusion injury[3–5]. Therefore, understanding of the molecular mechanisms in the ER stress and UPRs is critical for the management of patient with such diseases. Despite significant advances in understanding the UPRs, the pathway upstream to BiP is poorly understood. In particular, we have no definite ideas for how a broad range of stresses from pathophysiological conditions is interfaced with the events in the ER lumen[3,6]. For example, DNA double-strand break (DSB) responses may follow signaling pathways toward DNA-damage repair, while the DNA-damaged cells also set up separate mechanisms for cell death through UPRs[7,8]. In this scenario, DNA-damage repair is a nuclear event, but cell death processes occur manly in the cytoplasm. However, it is poorly understood how the DSB signals are transmitted to the ER in the cytoplasm[9].

Nascent polypeptide-associated complex α subunit (αNAC) and γ-taxilin (γTX) have been implicated in stress-induced apoptosis[7,10]. αNAC can bind growing nascent chains (NCs) emerging from the ribosome and modulate the action of signal-recognition particles, which transport the NCs into the ER lumen[11,12]. αNAC is a component of the ribosomal exit tunnel, providing a shield for NCs, and acting as a negative regulator of NC translocation into the ER. γTX is a member of the syntaxin-binding taxilin protein family, participating in the intracellular vesicle trafficking. The taxilin family, also including α and βTXs, can bind αNAC[13]. Therefore, αNAC and γTX may serve as cytoplasmic, extra-ER chaperons for the proper trafficking of nascent proteins. Notably, RNA interference of either of these proteins triggers UPRs leading to apoptotic cell death[7,10]. These results have raised the possibility that αNAC and γTX have a critical role in regulating the UPRs. Furthermore, the previous studies have shown that the αNAC/γTX pathway is strictly governed by the activity of glycogen synthase kinase 3β (GSK3β)[7,10]. However, the pathway further upstream to the GSK3β-αNAC/γTX axis remains unknown.

In this study, we provide the evidence that cytoplasmic ataxia telangiectasia mutated (ATM) protein is involved in stress-induced signal transduction responses. In this pathway, cytoplasmic ATM protein acts like a platform, where protein phosphatase 2A (PP2A) mediates dephosphorylation of AKT, thereby relaying the signals downwards to the GSK3β-αNAC/γTX axis and leading to mitochondria-dependent apoptosis in response to various forms of stress.

## Results

### The GSK3β/αNAC/γTX axis locates upstream of UPR signaling. Previous studies have shown that GSK3β mediates ER-stress

responses in hypoxic cells[7,10]. We found that camptothecin (CPT), ionizing radiation (IR), etoposide (Eto) downregulated Ser9-phosphorylated GSK3β (p-GSK3β), αNAC and γTX, and triggered UPRs, as evidenced by the upregulation of BiP and the activation of main ER-stress response sensors PERK, IRE1α, and ATF6 in HeLa S3 cells (Fig. 1a; Supplementary Fig. 1). Consistent with the notion that activated IRE1α oligomers mediate splicing of *XBP1* mRNA into *XBP1s* mRNA[14], XBP1s protein was upregulated (Fig. 1a). Thus, all the UPR branches are turned on the present experimental model. To monitor apoptotic cell death, we chose CHOP, p-JNK, Bax, and cleaved caspase-9 (c-casp-9) among many proteins involved in the ER-stress-induced cell death processes[3]. We found that the changes in the URP main branches were followed by the activation of JNK (p-JNK), CHOP, Bax and caspase-9 (c-casp-9), which paralleled the appearance of biochemical and morphological markers for apoptotic cell death (Fig. 1a–c). Furthermore, our previous studies using the same experiment models as in this study showed that siRNA-mediated αNAC or γTX depletion can trigger UPRs and accelerate cell death[7,10].

As expected, p-Chk2 (Thr68) and p-p53 (Ser15) were upregulated in CPT- or IR-treated HeLa S3 cells, confirming activation of these effectors downstream of ATM were activated under DNA damage (Supplementary Fig. 2a). GSK3β-mediated Tip60 phosphorylation has been implicated in the induction of apoptosis through the Puma/Bax axis[15], and Tip60-dependent p53 acetylation can induce apoptosis via increased mitochondrial membrane permeability[16,17]. These findings imply that Tip60 and Bax may be activated in the ER-stress-induced apoptotic pathway. We therefore tested this possibility in HeLa S3 cells treated with CPT or IR, and found that CPT and IR both induce Tip60 phosphorylation (Ser86) and Bax upregulation (Fig. 1a). By contrast, p-Tip60 was not upregulated in hypoxic cells (Supplementary Fig. 2b). Thus, the present data do not fully support the involvement of Tip60 in the ER-stress-induced apoptotic pathway.

It is of note that the kinetics of PERK, IRE1α and JNK activation after CPT treatment (peaks at ~6 h) was distinct from that observed after IR treatment (peaks at ~24–72 h) (Fig. 1a). The difference in kinetics may reflect the magnitude of ER-stress impacts on the cells. Consistent with this idea, the annexin-positive cell ratios were inversely correlated with the times required for PERK, IRE1α, and JNK activation to reach their peaks in HeLa S3 cells, with the ratios of ~12% at 6 h and ~77% at 24 h after CPT treatment (p-PERK, p-IRE1α, and p-JNK peaks at ~6 h); and the ratios of ~21% at 24 h, ~40% at 48 h, and ~61% at 72 h after IR treatment (peeks at ~24–72 h). Correlations between time courses of UPR-related protein activation and the cell's ultimate fate under ER stress have been suggested, with an emphasis on the critical timing of IRE1 activation and its termination in determining cell death fate[18]. Consistent with this hypothesis, the timing of IRE1α activation/termination as monitored with p-IRE1α protein levels was correlated with the annexin-positive cell fractions in cells under ER stress (Fig. 1a, b).

Then, we tested whether the activation of GSK3β is responsible for triggering UPRs in CPT- or IR-treated HeLa S3 cells. To this end, we used two distinct types of GSK3β-specific inhibitors, CHIR 99021 (CHIR) and lithium chloride (LiCl)[19]. We found that CHIR restored the expression levels of p-GSK3β, αNAC, and γTX, and suppressed the expression of cell death-related proteins (Fig. 1d). In addition, CHIR rescued CPT-treated cells from apoptotic cell death in a dose-dependent fashion (Fig. 1e, f). Similar results were obtained with LiCl (Fig. 1e, g, h). Concomitant siRNA ablation of GSK3β recovered the αNAC and γTX protein levels of CPT- or IR-treated cells, suggesting an indispensable role of active GSK3β in the signal transduction

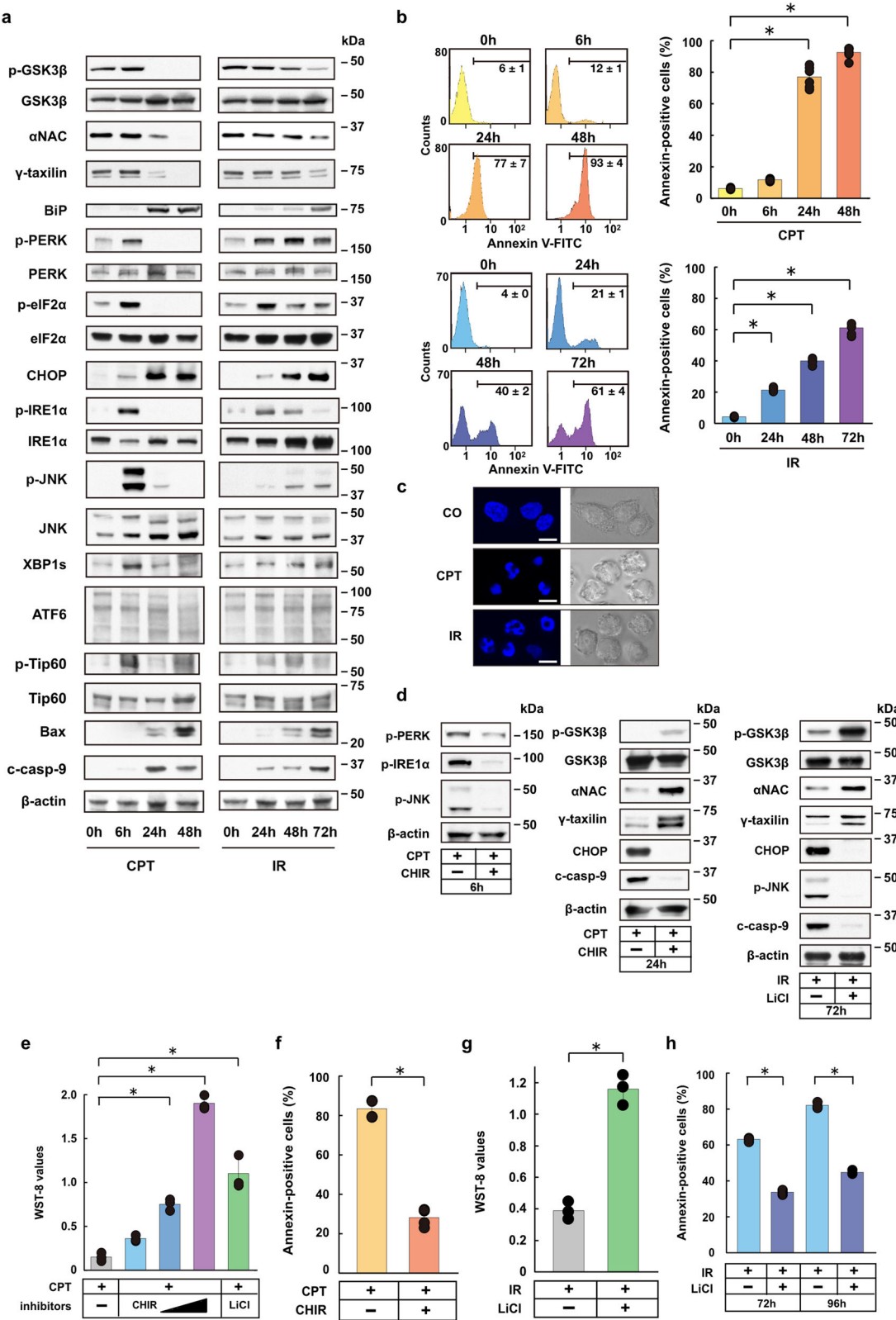

(Supplementary Fig. 3). Collectively, these results suggest that the GSK3β/αNAC/γTX axis locates upstream of UPR signaling.

**AKT is involved in GSK3β-mediated αNAC and γTX degradation**. The next question is how the GSK3β protein is activated under ER stress. It is well known that GSK3β can be regulated by

PI3K-dependent signaling that leads to the activation of AKT by phosphorylating the protein at Ser473 or Thr308 (p-AKT)[20]. Therefore, we assessed the expression levels of PI3K and p-AKT in HeLa S3 cells that were treated with CPT or IR, or in hypoxic cells. AKT was phosphorylated at Ser473, but not at Thr308 in control HeLa S3 cells; however, p-AKT was downregulated after CPT, IR, or hypoxic treatment (Fig. 2a). In contrast, protein levels

**Fig. 1 GSK3β-dependent ER-stress response pathway via αNAC and γTX degradation. a** Western blot analysis for GSK3β, αNAC, γTX, and ER-stress response-related proteins in camptothecin (CPT, 1 μM)- or ionizing radiation (IR, 20 Gy)-treated HeLa S3 cells. **b** Fluorescence-activated cell sorter (FACS) analysis shows annexin-positive ratios of HeLa S3 cells before (0 h) and varying times (6–72 h) after CPT or IR treatments. Horizontal lines in FACS histogram indicate annexin-positive cell fractions (%). Bars, mean ± s.e.m.; n = 6 independent experiments; *P < 0.001, Tukey–Kramer test. **c** Confocal microscopy shows fragmented nuclei (DAPI staining, left panel) of CPT- or IR-treated HeLa S3 cells. Right panel, phase-contrast micrographs. Scale bar = 10 μm. **d** Western blot analysis shows protein expression levels of αNAC, γTX UPRs and ER-stress response-related death signals (CHOP, p-JNK, c-caspase-9) in CPT- (left and center panels) or IR-treated (right panel) HeLa S3 cells in the presence or absence of GSK3β-specific inhibitor (CHIR or LiCl). **e** Bar graph shows recovery (increased WST-8 values) by CHIR or LiCl from CPT-induced cell death in HeLa S3 cells. Bars, mean ± s.e.m.; n = 3 independent experiments; *P < 0.001, Tukey–Kramer test. **f** Bar graph shows inhibition by CHIR of CPT-induced cell death (annexin-positive cells) in HeLa S3 cells. Bars, mean ± s.e.m; n = 6 independent experiments; *P < 0.001, t test. **g** Bar graph shows increased viability of IR-treated HeLa S3 cells after LiCl treatment. Bars, mean ± s.e.m.; n = 3 independent experiments; *P < 0.001, t test. **h** Bar graph shows inhibition by LiCl of IR-induced cell death in HeLa S3 cells. Bars, mean ± s.e.m.; n = 4 (72 h) or 3 (96 h) independent experiments; P < 0.001, t test.

of PI3K complex were relatively constant after such treatments. Notably, the PI3K-specific inhibitor LY294002 did not affect the expression levels of p-AKT or total AKT, and the protein levels of phosphorylated as well as total GSK3β, αNAC, and γTX remained mainly constant after the PI3K inhibition (Fig. 2b). However, HeLa S3 cells that were treated with the highly selective pan-AKT inhibitor MK2206 or perifosine, exhibited expression kinetics of AKT, GSK3β, αNAC, γTX, CHOP, JNK, and c-caspase-9, similar to those of cells treated with CPT, IR, or of hypoxic cells (Fig. 2b vs. Fig. 1a)[7,10]. Furthermore, the AKT inhibitors accelerated apoptotic cell death, whereas the PI3K inhibitor LY294002 did not (Fig. 2c), suggesting that AKT, but not PI3K is involved in ER-stress responses leading to apoptotic cell death. PKC signaling could be another potential effector upstream of AKT in the ER-stress-induced pro-apoptotic pathway, since some types of PKC family proteins can phosphorylate AKT and GSK3β[21]. To test this possibility, we asked whether PKC inhibition could activate AKT/GSK3β/αNAC/γTX-dependent apoptotic pathway. We found that PKCα and PKCδ expression levels are not changed in cells under ER stress caused by CPT, IR, or hypoxia (Supplementary Fig. 4a). Moreover, a pan-PKC inhibitor sotrastaurin did not down-regulate p-AKT, p-GSK3β, αNAC, or γTX, and the inhibition did not accelerate apoptotic cell death (Supplementary Fig. 4b, c). These data indicate that the PKC signaling is unlikely to be involved in the pro-apoptotic pathway under ER stress.

To further evaluate the role of GSK3β in the AKT inhibition-induced degradation of αNAC or γTX and apoptotic cell death, HeLa S3 cells were treated with perifosine in the presence or absence of CHIR, and then αNAC and γTX protein expression levels and apoptotic cell ratios of these cells were monitored. We found that the GSK inhibitor restores αNAC and γTX protein expression levels, and suppresses caspase-9 activation and apoptotic cell death of perifosine-treated cells (Fig. 2d, e), suggesting a pivotal role of GSK3β in the AKT inhibition-induced apoptotic pathway. Several feedback mechanisms are inherent in the PI3K/AKT signaling pathway, typically via the downstream effectors of AKT[20]. Therefore, it is plausible that the GSK3β inhibition by CHIR or LiCl in cells under ER stress may feedback regulate the AKT activity. To test this possibility, we analyzed p-AKT expression levels in CPT or IR-treated HeLa S3 cells in the presence or absence of a GSK3β inhibitor (LiCl or CHIR). We found that the GSK3β inhibition upregulates p-AKT in CPT- or IR-treated cells (Fig. 2f). These results suggest that feedback-regulated AKT activation contributes to the rescue of ER-stressed cells from apoptotic cell death.

To exclude potential off-target effects of the AKT inhibitors, AKT was separately depleted by RNA interference in HeLa S3 cells. We found that the siRNA-mediated AKT depletion substantially suppresses GSK3β phosphorylation, downregulates αNAC and γTX, and induces CHOP and JNK activation (Fig. 2g). However, these changes were not observed in cells treated with a

control siRNA or in cells that were mock treated. Furthermore, the AKT siRNA, but not mock treatment or control siRNA, significantly accelerated apoptotic cell death (Fig. 2h). Therefore, the siRNA-mediated AKT inhibition triggered ER-stress responses similar to those caused by the pharmacological AKT inhibitor. These results support the notion that AKT acts as an upstream regulator of GSK3β activation leading to the αNAC and γTX depletion-mediated ER-stress responses and cell death pathway.

**ATM protein allows for an interplay of AKT and GSK3β.** We hypothesized that ATM protein may serve as a platform for AKT inactivation events in cells that are committed to cell death under ER stress. The reason is twofold: ATM is involved in signaling processes that occur in the cytoplasm[22,23]. Second, cross talk has been found between ATM and AKT and between ATM and HIF[24,25]. To test this hypothesis, we first explored the subcellular localization of ATM, AKT, and GSK3β proteins in HeLa S3 cells. Confocal microscopy of untreated HeLa S3 cells showed that ATM is uniformly distributed in both the cytoplasm and nucleus, except for the nucleoli (Fig. 3a). By contrast, p-AKT localizes predominantly to the cytoplasm. Thus, the p-AKT colocalizes with the ATM in the cytoplasm and to a lesser extent within the nucleus. Western blot analysis of fractionated cell lysates pro-vided data consistent with the findings by confocal microscopy (Fig. 3b). To further interrogate the main interaction site of ATM and p-AKT proteins, we performed proximity ligation assay (PLA). We found that PLA fluorescence puncta localize to both the cytoplasm and nucleus of HeLa S3 cells, with most PLA signals detected in the perinuclear region of the cytoplasm or in the peripheral part of the nucleoplasm (Fig. 3c). Considering the 3D confocal PLA imaging with z-stacking and maximum inten-sity projection (MIP), where the perinuclear cytoplasm and the nuclear periphery may overlap on stacked images, these results suggest that the interactions between ATM and p-AKT proteins occur predominantly in the perinuclear region of the cytoplasm.

The physical interactions of ATM and p-GSK3β or p-AKT proteins were further evaluated by performing immunoprecipita-tion assays using 293T cells that were transfected with FLAG-tagged ATM 48 h prior to varying ER-stress-inducing treatments. Intriguingly, the kinetics of total AKT protein levels is distinct between HeLa S3 (Fig. 2a) and 293T cells (Fig. 3d), with sharp increases in the AKT level being observed 5–24 h after ER-stress induction in 293T cells, as opposed to HeLa S3 cells without such upregulation kinetics of total AKT, but the protein rather unchanged throughout the time courses. Nevertheless, p-AKT was similarly downregulated in both the cell lines that were treated with different types of ER-stress inducers. The observed difference in the AKT expression levels between these cell lines may be due to a possible difference in intrinsic growth potential between these cell lines. In general, AKT activation promotes cell

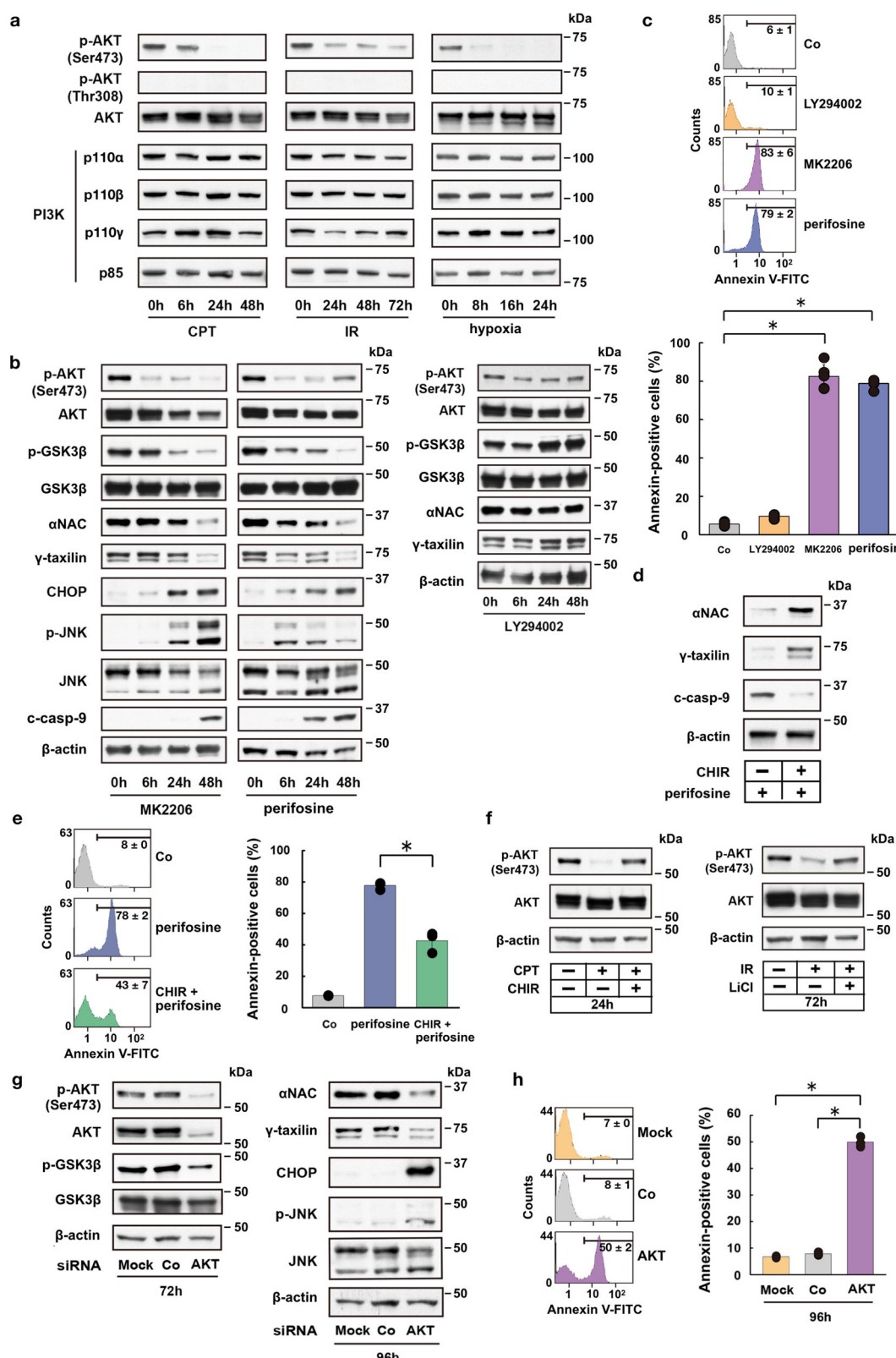

survival, proliferation, and growth[20]. Therefore, we speculate that the excess of unphosphorylated over phosphorylated AKT in non-stressed HeLa S3 cells (Fig. 2a) reflects cellular machinery intrinsically programmed against the cell's propensity for high proliferation rates[26], opposing to 293T cells, in which phosphorylated AKT is predominant over unphosphorylated AKT (Fig. 3d). Upon ER stress, 293T cells may respond the stress by

means of decreasing phosphorylated AKT and simultaneously by increasing unphosphorylated AKT in order to effectively suppress the cell proliferation, while HeLa S3 cells can achieve this by downregulating phosphorylated AKT without the need for upregulating unphosphorylated AKT.

In control 293T cells, AKT protein co-immunoprecipitated with FLAG-tagged ATM was predominantly phosphorylated

**Fig. 2 AKT modulates GSK3β-dependent ER-stress response pathway. a** Western blot analysis shows kinetic changes of Ser473 phosphorylated AKT (p-AKT) and PI3K component (p110α, p110β, p110γ, p85) protein levels in CPT- or IR-treated, or hypoxic HeLa S3 cells. **b** Western blot analysis shows kinetic changes of p-AKT (Ser473), p-GSK3β (Ser9), αNAC, γTX, and ER-stress response-related proteins in HeLa S3 cells after addition of AKT (MK2206 or perifosine) or PI3K (LY294002) inhibitors. **c** FACS analysis shows accelerated apoptotic cell death by AKT inhibition (MK2206 or perifosine), but not by PI3K inhibition (LY294002) in HeLa S3 cells. Bars, mean ± s.e.m.; $n = 6$ independent experiments; *$P < 0.001$, Tukey–Kramer test. **d** Western blot analysis for αNAC, γTX and c-casp-9 following GSK3β inhibition (CHIR) in perifosine-treated HeLa S3 cells. **e** FACS analysis shows suppression of apoptotic cell death following GSK3β inhibition (CHIR) in perifosine-treated HeLa S3 cells. Bars, mean ± s.e.m.; $n = 3$ independent experiments; *$P < 0.001$, Tukey–Kramer test. **f** Western blot analysis shows feedback regulation of p-AKT after GSK3β inhibition (CHIR or LiCl) in CPT- or IR-treated HeLa S3 cells. **g** Western blot analysis shows AKT siRNA-induced suppression of p-GSK3β (72 h), αNAC (96 h), and γTX (96 h) expression, and induction of CHOP and p-JNK expression (96 h) in HeLa S3 cells. **h** FACS analysis shows AKT siRNA-induced apoptotic cell death in HeLa S3 cells. Bars, mean ± s.e.m.; $n = 3$ independent experiments; *$P < 0.001$, Tukey–Kramer test.

(activated; Fig. 3e). However, ATM-associated AKT proteins were largely unphosphorylated (inactivated) in cells treated with CPT or IR at 24 h after treatment; and in cells under hypoxic conditions for 5 h. Both phosphorylated (inactivated) and unphosphorylated (activated) GSK3β proteins were also co-immunoprecipitated with FLAG-tagged ATM in control cells, whereas the phosphorylated proteins had almost disappeared in the FLAG-tagged ATM precipitates from CPT or IR-treated, and hypoxic cells (Fig. 3f).

The binding sites for AKT and GSK3β on the ATM protein have not been identified. However, results from confocal microscopy (Fig. 3a), PLA analysis (Fig. 3c) and immunoprecipitation assays (Fig. 3e, f) imply that cytoplasmic ATM can recruit AKT and GSK3β proteins and then AKT phosphorylates GSK3β on ATM[27]. Mutations of the ATM protein at autophosphorylation sites and of the C-terminal kinase domain can greatly influence the activity of the protein[23,28]. However, none of well-known ATM mutations of S1981A or D2870A/N2875K (kinase-dead, KD), or possible mutations at R533A, K2117A, or K2992A/S2996A that were deduced based on the results from mass spectroscopic analysis, affected the binding and phosphorylation states of the endogenous AKT proteins (Fig. 3g, h; and Supplementary Table 1).

A potent ATM-specific inhibitor KU-60019 can attenuate AKT phosphorylation at Ser473 and interfere with cell growth in human glioma cells and enhance apoptotic cell death and alleviate senescence[29–31]. Accordingly, we tested the possibility that KU-60019 could modulate the ER-stress-induced death signaling pathway. KU-60019 treatment caused changes in protein expression levels of ATM, AKT, GSK3β, and downstream UPR-related proteins, which mimicked those in HeLa S3 cells undergoing ER-stress responses (Fig. 3i vs. Fig. 1a). In addition, KU-60019 treatment accelerated cell death in a dose-dependent fashion (Fig. 3j). Furthermore, ATM ablation by RNA interference causes downregulation of p-AKT, p-GSK3β, αNAC, and γTX, and triggers UPRs such as CHOP and JNK activation, findings that are again reminiscent of the changes caused by CPT, IR, hypoxic stress, or KU-60019 (Supplementary Fig. 5a vs. Figs. 1a and 3i)[7,10]. These data further define the link between ATM and AKT in ER-stress responses. However, apoptosis induction by the ATM ablation was less effective compared with the other ER-stress inducers tested: ~33% for ATM ablation vs. ~93% for CPT, ~61% for IR, and ~73% for KU-60019 in HeLa S3 cells (Supplementary Fig. 5b vs. Figs. 1b and 3j). These data imply differential cell death impacts between the pharmacological inhibition of ATM and the siRNA-mediated ATM protein ablation. The degradation of ATM proteins detected in KU-60019-treated cells and cells undergoing ER-stress responses (Fig. 3i; Supplementary Fig. 6) appears to be the consequence of caspase activation in late stages of cell death processes since a pan-caspase inhibitor Z-VAD-fmk completely suppressed ATM protein degradation; however, the caspase inhibitor did not revive

the AKT or GSK3β protein levels in KU-60019-treated HeLa S3 cells (Fig. 3k).

As expected, DNA-damaging agents (CPT, IR and etoposide) induced ATM phosphorylation at Ser1981 during early time courses, whereas KU-60019 treatment was not associated with this hallmark of DNA-damage-induced ATM activation (Fig. 3i; Supplementary Fig. 7), instead provoking caspase-dependent protein degradation after ER-stress induction (Fig. 3i, k; Supplementary Fig. 6). Therefore, the DNA-damaging agents and KU-60019 exerted opposing effects on the ATM activity. Nevertheless, both the treatment types similarly enhanced UPRs and JNK phosphorylation, and comparably induced apoptotic cell death (Figs. 1a, b and 3i, j). These results suggest that ER-stress responses leading to cell death can occur without activating ATM protein, at least not necessarily requiring the canonical phosphorylation at Ser1981 unlike in DNA-damage responses. In addition, DNA-damage-induced ATM activation and the subsequent recruitment of the activated protein to the damaged DNA sites occurs at early stages (<~1 h) of DNA-damage responses in the nucleus[32], whereas the UPRs become evident in the cytoplasm at later (>~6 h) stages after ER-stress induction by DNA-damaging agents or KU-60019 (Figs. 1a and 3i). Therefore, the early DNA repair events in the nucleus are seeking to rescue the cell from cell death by means of activating the nuclear ATM[24]; however, the UPR events occurring at later stages in the cytoplasm may use the cytoplasmic ATM as a cofactor or a platform, directing the cell to cell death processes when the earlier nuclear events cannot ameliorate the damaged states in the cell.

**PP2A plays a crucial role in regulating AKT in stress.** AKT phosphorylation site Ser473 is not a consensus S/T-Q ATM kinase motif, and the AKT phosphorylation process is protein phosphatase okadaic acid-sensitive[29]. These results suggest that ATM may not directly phosphorylate AKT; instead, ATM may contribute to the AKT-GSK3β signaling pathway by indirectly regulating AKT phosphorylation in a protein phosphatase-dependent fashion. Previous studies have demonstrated the involvement of protein phosphatases in the regulation of ATM activity[33,34]. Of these phosphatases, protein phosphatase 2A (PP2A) is also involved in the regulation of AKT phosphorylation[20]. Therefore, we tested this notion by analyzing for PP2A proteins in FLAG-tagged ATM precipitates from CPT- or IR-treated, and hypoxic 293T cells. The FLAG-tagged ATM precipitates recovered from control 293T cells contained trace amounts of PP2A proteins (Fig. 4a). However, CPT, IR, and hypoxia all increased the levels of these proteins in the FLAG-tagged ATM precipitates obtained at 5 or 24 h after the treatments. In contrast, PP1α and PHLPP, which are also responsible for the regulation of AKT phosphorylation[35–37], were not detected in the precipitates (Fig. 4a, b). Moreover, increased amounts of PP2A phosphatase were detected in the FLAG-tagged, mutated ATM precipitates from IR-treated and/or hypoxic HeLa

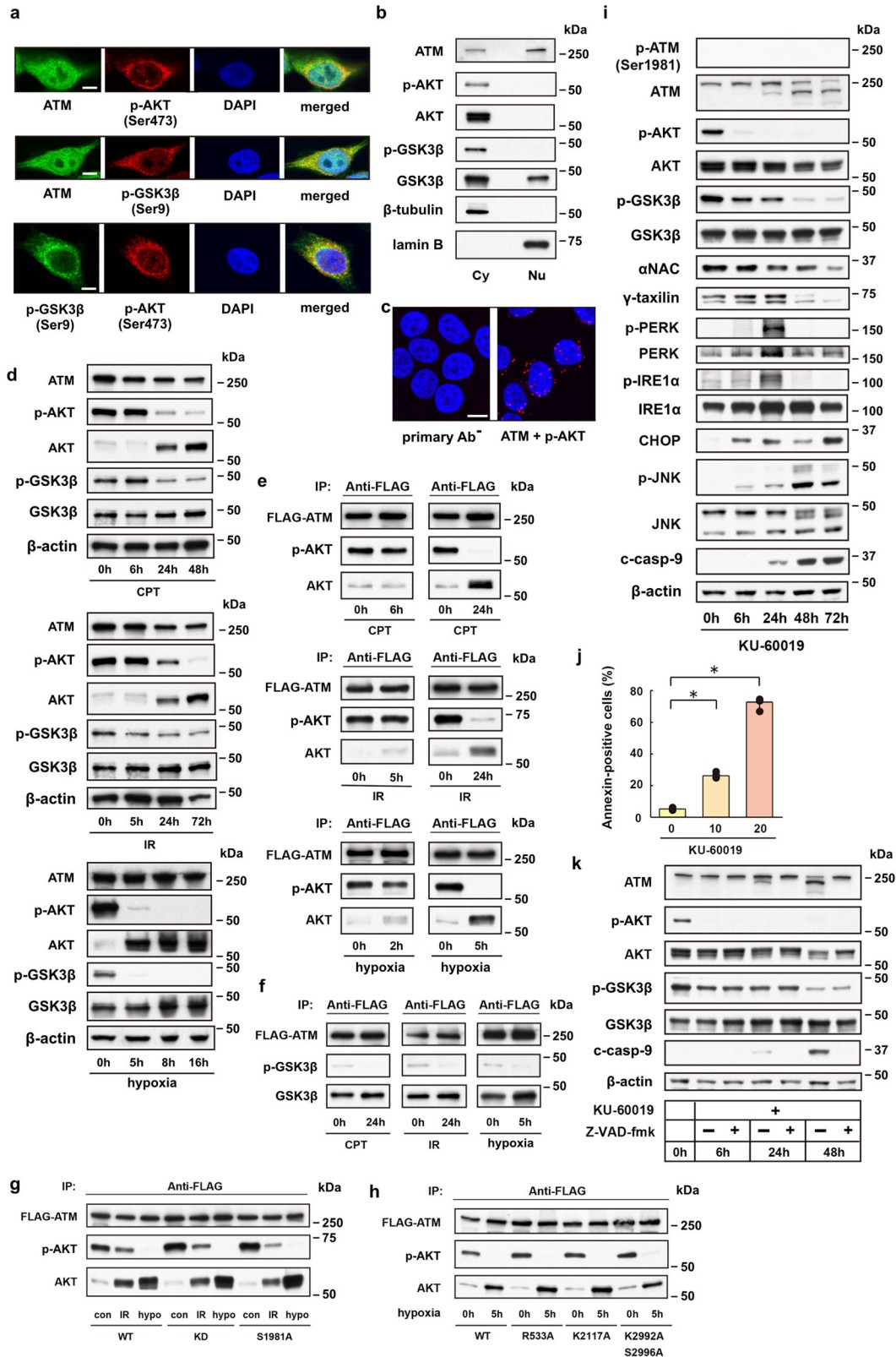

S3 cells, indicating that these mutations do not affect the interaction between the mutated ATM proteins and PP2A phosphatases (Supplementary Fig. 8).

We extended these findings to investigate the effect of phosphatase inhibitors on ATM/AKT-dependent cell death pathways under ER stress. We found that the phosphatase inhibitor calyculin A blocked the binding of PP2A-A, PP2A-B,

and PP2A-C to the ATM protein in CPT- or IR-treated cells, as well as in hypoxic cells (Fig. 4c). The binding of PP2A proteins to ATM protein was also inhibited by a structurally unrelated protein phosphatase inhibitor okadaic acid. More importantly, the protein phosphatase inhibitors efficiently suppressed phosphatase activity bound to the FLAG-tagged ATM (Fig. 4d) and rescued the cells from apoptotic cell death caused by CPT-, IR-,

**Fig. 3 Cytoplasmic ATM serves as a platform on which AKT protein is inactivated under ER stress. a** Fluorescence confocal microscopy shows subcellular localization of ATM, p-AKT and p-GSK3β in HeLa S3 cells. Scale bar = 5 µm. **b** Western blot analysis for AKT, p-AKT, GSK3β and p-GSK3β proteins in cytoplasmic (Cy) or nuclear (Nu) cell lysates from HeLa S3 cells. **c** Proximity ligation assay (PLA) reveals multiple PLA puncta in the cytoplasm and to lesser extent in the nuclear periphery of HeLa S3 cells. PLA without primary antibodies (primary Ab⁻) or with ATM- and p-AKT-specific antibodies (ATM + p-AKT). Scale bar = 10 µm. **d** Western blot analysis for 293T cells shows time-dependent downregulation of p-AKT and p-GSK3β proteins after treatment with CPT (1 µM) or IR (20 Gy), or under hypoxic stress. **e** Immunoprecipitation assay shows depletion of p-AKT in FLAG-tagged ATM precipitates from 293T cells treated with CPT (24 h) or IR (24 h), or from cells under hypoxia (5 h). **f** Immunoprecipitation assay for GSK3β and p-GSK3β proteins in FLAG-tagged ATM precipitates from 293T cells after treatment with CPT or IR, or from cells under hypoxia. **g** Immunoprecipitation assay for AKT and p-AKT proteins in FLAG-tagged wild-type (WT) or mutant (KD or S1981A) ATM precipitates before (con) and after ER stress (IR or hypo). **h** Immunoprecipitation assay for AKT and p-AKT proteins in FLAG-tagged wild-type (WT) or mutant (R533A, K2117A, or K2992A/S2996A; Supplementary Table 1) ATM precipitates from control (0 h) and hypoxic (5 h) 293 T cells. **i** Western blot analysis shows ER-stress response-mimicking changes in protein expression levels caused by the ATM-specific inhibitor KU-60019 in HeLa S3 cells. **j** Bar graph shows dose-dependent (0, 10, or 20 µM) enhancement by KU-60019 of apoptotic cell death. Bars, mean ± s.e.m.; n = 5 independent experiments; *P < 0.001, Tukey–Kramer test. **k** Western blot analysis shows caspase-dependent (Z-VAD-fmk-sensitive) degradation of ATM in KU-60019 (20 µM)-treated HeLa S3 cells, opposing to caspase-independent (Z-VAD-fmk-insensitive) downregulation of p-AKT and p-GSK3β.

---

or hypoxia-induced ER stress (Fig. 4e). The FLAG-tagged ATM precipitates from KU-60019-treated 293T cells contained unphosphorylated, but not phosphorylated AKT and trace amounts of PP2A proteins, suggesting that the changes are PP2A-independent (Fig. 4f, g). A possible explanation may be that KU-60019 directly prevents phosphorylation at Ser473[29].

## Discussion

ER-stress response-induced cell death mechanisms have been extensively studied, but the pathway upstream of the signaling events in the ER lumen has remained elusive[1]. In this study, we have identified the ATM-AKT-GSK3β-αNAC/γTX signaling axis, which, unlike DNA-damage responses in the nucleus, is activated in the cytoplasm (Fig. 4h). In this pathway, cytoplasmic ATM protein acts like a platform, where AKT protein carries death-signal passengers in a phosphatase-dependent fashion like a train bound for cell death.

Recently, ER stress has been suggested to be involved in the pathogenesis of several human diseases, including cancer, autoimmune diseases, diabetes, and brain ischemia[3]. Many facets of the ATM-AKT-GSK3β-αNAC/γTX signaling axis in hypoxic conditions are involved in pathogenesis of neurodegenerative diseases[38,39]. Notably, downregulation of αNAC/γTX occurs in the brains of patients with Alzheimer's disease[10]. Therefore, better understanding of the ATM-AKT-GSK3β-αNAC/γTX signaling axis may provide new insights into the pathogenesis of neurodegenerative diseases and ER- stress-related human diseases, and could be valuable in evolving novel treatment strategies.

## Methods

**Cell culture.** HeLa S3 human cervical cancer cells (ATCC) and 293T human embryonic kidney cells (ATCC) were cultured in DMEM supplemented with 10% fetal bovine serum. The cells were treated with CPT (1 µM, Sigma), IR (160 kV, 1.0 mm Al, 20 Gy), or Eto (10 µM, Merck Millipore) and allowed to continue growing for indicated times. For hypoxic stress, the cells were cultured under hypoxic conditions (<1% pO₂ and 5% pCO₂) using an anaerobic culture kit (Mitsubishi Gas Chemicals). The system provides a hypoxic condition without affecting the pH of the medium. We confirmed that the cancer cell lines used in this study are not listed in the database of commonly misidentified cell lines (ICLAC). None of the cell lines used were authenticated. We did not test for mycoplasma contamination of the cell lines used.

**Inhibitors.** Inhibitors used in this study were as follows: CHIR 99021 (5, 15, or 30 µM, Stemgent) and LiCl (40 mM, Sigma) for GSK3β inhibition; LY294002 (50 µM, Cell Signaling) for PI3K inhibition; MK2206 (20 µM, Selleckchem) and perifosine (25 µM, Selleckchem) for AKT inhibition; KU-60019 (10 or 20 µM, Selleckchem) for ATM inhibition; sotrastaurin (10 µM, Selleckchem) for pan-PKC inhibition; z-VAD-fmk (100 µM, BD Pharmingen) for pan-caspase inhibition; calyculin A (5 or 10 nM, Melck Millipore) and okadaic acid (1 µM, Melck Millipore) for protein phosphatase inhibition.

**Western blotting.** Cells were washed in ice-cold phosphate-buffered saline (PBS), and lysed in a buffer containing 10 mM HEPES (pH 7.9), 1.5 mM MgCl₂, 10 mM KCl, 0.5 mM DTT, and protease inhibitor cocktail (Roche). Equal amounts of proteins (~40 µg) were then analyzed on a 7.5 or 10% sodium dodecylsulphate (SDS) polyacrylamide gel, or on a 4–15% SDS polyacrylamide gradient gel.

**Immunoprecipitation assay.** A FLAG-tagged ATM was transfected into 293T cells 48 h prior to varying treatments. After indicated treatment periods, the cells were collected, washed in ice-cold PBS, and lysed in Pierce IP Buffer (Thermo) on ice for 5 min. The lysates were centrifuged at 13,000 × g for 10 min, and the supernatants were stored at −80 °C until use. Immunoprecipitation assay was performed by using a FLAG immunoprecipitation kit (FLAGIPT1, Sigma), and the precipitates were eluted with a sample buffer, and then co-immunoprecipitated proteins were analyzed on an SDS polyacrylamide gel as described above.

**Immunofluorescence microscopy.** Cells were washed with PBS and fixed with 4% paraformaldehyde in PBS for 20 min, and were then permeabilized with 0.2% Triton X-100 in PBS for 15 min at room temperature. Blocking was performed with 5% goat serum in PBS for 1 h at room temperature. Incubation with the primary antibody was performed overnight at 4 °C. Proteins were detected after incubating the cells with secondary antibodies for 2 h at room temperature. Visualization of nuclei was achieved by incubating cells with DAPI (1 µg/ml) for 10 min at room temperature. Immunofluorescence visualization was carried out under a TCS SP2 AOBS confocal microscope (Leica).

**Proximity ligation assay (PLA).** PLA was performed using a commercially available kit (Duolink in Situ Starter Set ORANGE, Sigma-Aldrich) and according to the manufacturer's instructions. After fixing cells with 4% paraformaldehyde in PBS on a slide, the cells were permeabilized with 0.2% Triton X-100 in PBS for 15 min at room temperature. After blocking for 30 min at 37 °C in a humidity chamber, the cells were stained with the primary antibodies for ATM and p-AKT proteins overnight at 4 °C. Then, the cells were stained with anti-mouse PLA PLUS (for p-AKT) and anti-rabbit PLA MINUS (for ATM) secondary antibodies (PLA probes) for 60 min at 37 °C in a pre-heated humidity chamber. Subsequently, the samples were subjected to a ligation reaction for hybridization of the two oligonucleotides in the PLA probes to the circularization oligonucleotides for 30 min at 37 °C, followed by PCR amplification for 100 min at 37 °C, both performed in a pre-heated humidified chamber. Finally, the slide was mounted with a coverslip using a mounting medium containing DAPI, followed by 3D confocal imaging. The cells were imaged with z-stacks of 0.6-µm thickness to detect all PLA puncta in the cells.

**Antibodies.** The primary antibodies used for western blotting included: γ-taxilin (Santa Cruz Biotechnology, sc-393610, used at a concentration of 1:1000), αNAC (Abnova, H00342538, 1:1000), ATF6 (IMGENEX, IMG-273, 1:200), BiP (KDEL; Stressgen, spA-827, 1:200), CHOP (Cell Signaling, 2895, 1: 1000), PERK (Santa Cruz Biotechnology, sc-13073, 1:200), phospho-PERK (Santa Cruz Biotechnology, sc-32577, 1:200), IRE1α (Cell Signaling, 3294, 1:1000), phospho-IRE1α (NOVUS, NB100-2323, 1:1000; abcam, ab124945, 1:1000), cleaved caspase-9 (Cell Signaling, 9505, 1:500), elF2α (Cell Signaling, 9722, 1:1000), phospho-elF2α (Cell Signaling, 9721, 1:500), GSK3β (Cell Signaling, 9315, 1:2000), phospho-GSK3β (Ser9, Cell Signaling, 9336, 1:1000), JNK (Cell Signaling, 9252, 1:1000), phospho-JNK (Cell Signaling, 4668, 1:500), Bax (Santa Cruz Biotechnology, sc-748, 1:200), p53 (Merck Millipore, Ab-6, 1:1000), phospho-p53 (Ser15, DNA-damage antibody sampler kit, Cell Signaling, 9947, 1:500), Chk2 (Cell Signaling, 2662, 1:2000), phospho-Chk2 (DNA-damage antibody sampler kit; Cell Signaling, 9947, 1:1000), Tip60 (Cell Signaling, 12058, 1:1000), phospho-Tip60 (Ser86, Abcam, ab73207, 1:500), XBP1s

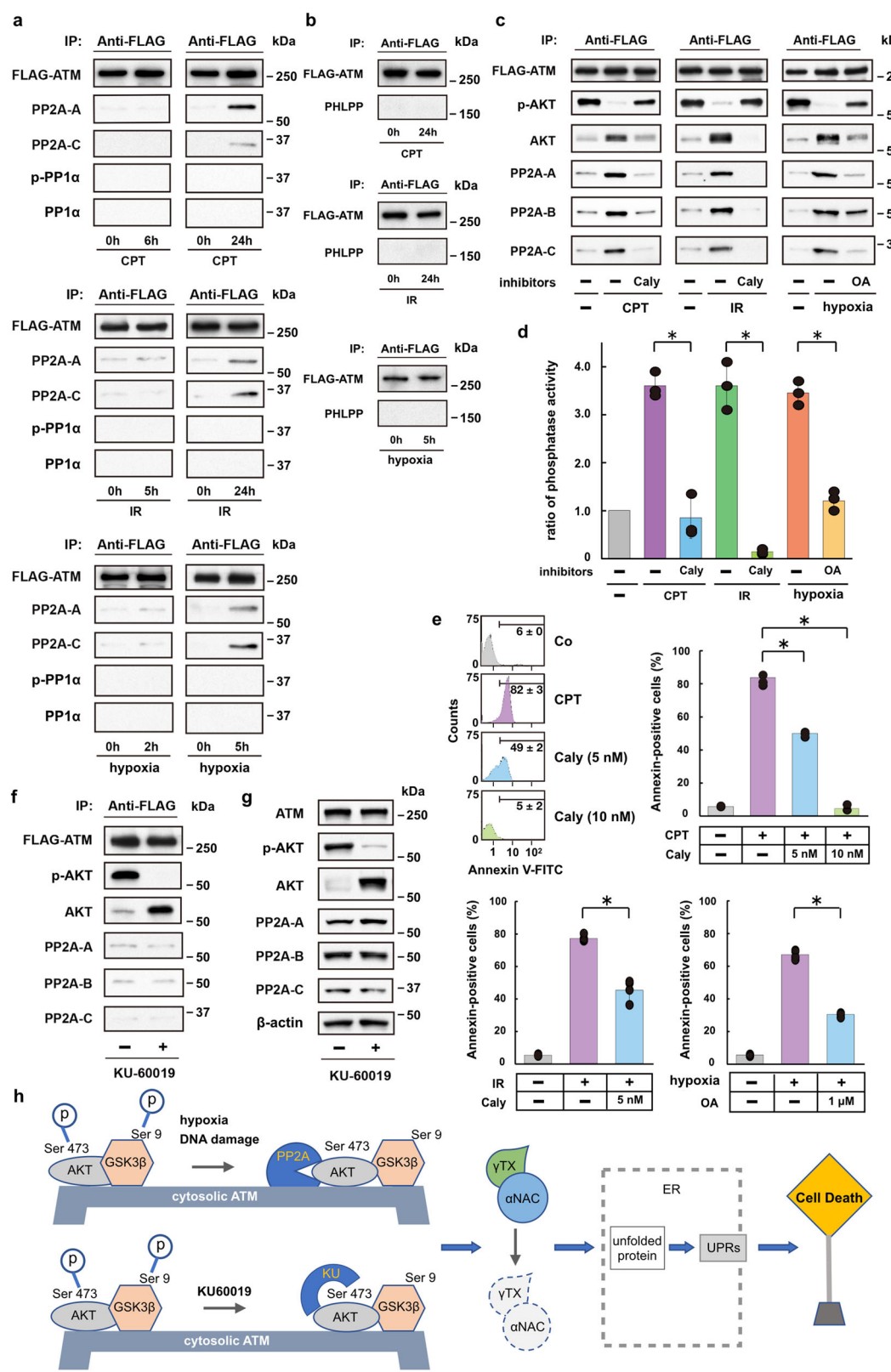

(Santa Cruz Biotechnology, sc-7160, 1:200), PI3K (Cell Signaling, p110α, 4249, 1:1000; p110β, 3011, 1:1000; p110γ, 5405, 1:1000; p85, 4257, 1:1000), ATM (Cell Signaling, 2873, 1:2000), phospho-ATM (Ser1981; Cell Signaling, 5883, 1:500), AKT (Cell Signaling, 4691, 1:2000), phospho-AKT (Ser473, Cell Signaling, 4060, 1:500), phosphor-AKT (Ser308, Cell Signaling, 13038, 1:500), PKCα (Cell Signaling, 2056, 1:1000), PKCδ (Cell Signaling, 9616, 1:1000), PP2A-A (Cell Signaling, 2041, 1:1000), PP2A-B (Cell Signaling, 2290, 1:1000), PP2A-C (Cell Signaling, 2259, 1:1000), PP1α (Cell Signaling, 2582, 1:500), phospho-PP1α (Thr320; Cell Signaling, 2581, 1:500), PHLPP (Bethyl, A300-661A, 1:1000), FLAG (Sigma, F7425, 1:1000), β-tubulin (Cell Signaling, 2146, 1:1000), lamin B (Cell Signaling, 12586, 1:1000), β-actin (Sigma, A3853, 1:1000). The secondary antibodies used for Western blotting included anti-rabbit (Cell Signaling, 7074, 1:1000) and anti-mouse (Cell Signaling, 7076, 1:1000) IgG antibodies. The primary antibodies used for immunofluorescence microscopy included ATM (Merck Millipore, 07-1286, 1:100), p-AKT (Santa Cruz Biotechnology, sc-514032, 1:100), and phospho-GSK3β (Thermo Fisher, MA5-14873, 1:100; Santa Cruz Biotechnology, 373800, 1:100).

**Fig. 4 Protein phosphatase PP2A regulates AKT on the cytoplasmic ATM under ER stress directing cell death. a, b** Immunoprecipitation assay for PP2A (PP2A-A and PP2A-C), p-PP1α and PP1α **a**, and PHLPP phosphatases **b** in FLAG-tagged ATM precipitates from CPT- or IR-treated, or hypoxic 293T cells **c** Immunoprecipitation assay for PP2A phosphatase (PP2A-A, PP2A-B, and PP2A-C) in FLAG-tagged ATM precipitates from CPT- or IR-treated, or hypoxic 293T cells following treatment with PP2A- and PP1-specific inhibitors (calyculin, Caly) or okadaic acid (OA). **d** Bar graph shows phosphatase activity ratios in FLAG-tagged ATP precipitates from CPT- or IR-treated, or hypoxic 293T cells, in the presence or absence of phosphatase inhibitor (Caly or OA). Bars, mean ± s.e.m.; $n = 3$ independent experiments; $P < 0.001$, Tukey–Kramer test. **e** Bar graphs from FACS analysis show annexin-positive ratios of HeLa S3 cells under ER stress (CPT, IR, or hypoxia) in the presence or absence of phosphatase inhibitor (Caly or OA). Bars, mean ± s.e.m.; $n = 4$ (CPT, IR and hypoxia) independent experiments; $*P < 0.001$, Tukey–Kramer test. **f, g** Immunoprecipitation assay **f** and western blot analysis **g** for p-AKT, AKT, and PP2A phosphatase (PP2A-A, PP2A-B, and PP2A-C) in FLAG-tagged ATM precipitates and total cell lysates, respectively, from 293T cells following treatment with the ATM-specific inhibitor KU-60019 (20 μM). **h** Schematic drawing of the proposed molecular mechanisms for apoptotic cell death pathway via PP2A/AKT-dependent GSK3β activation, which is carried out on the cytosolic ATM platform under ER-stress. The ATM-specific inhibitor KU-60019 can also cause GSK3β-dependent degradation of αNAC and γTX and the following UPRs and eventual cell death. However, this pathway is PP2A-independent; probably the inhibitor may directly target AKT protein.

---

The secondary antibodies used for immunofluorescence microscopy included FITC (Invitrogen, F2765, 1:400) and Cy3 (GE Healthcare, PA 43009, 1:2500). Each primary antibodies were validated on the manufactures' websites and/or relevant citations.

**RNA interference.** Oligonucleotides corresponding to human ATM (5′-GCAAAG CCCUAGUAACAUA-3′), GSK3β (5′-CCACAAGAAGUCAGCUAUATT-3′), and AKT(5′-GCUACUUCCUCCUCAAGAA-3′) were transfected into HeLa S3 cells using Lipofectamine RNAiMax (Invitrogen) according to the manufacturer's instruction. The effect of RNA interference was measured 48–96 h after the transfection. AllStar Negative Control siRNA (QIAGEN) was used as a control.

**Assessment of apoptosis and cell viability.** Apoptosis was assessed after incubating cells with Annexin V-FITC (Sigma) at room temperature for 10 min. Cells positive for Annexin were analyzed by FACS scan (Epics XL, Beckmann Coulter). In total, $10^4$ cells for each treatment/non-treatment group were used for FACS analyses. Cell viability was measured by a modified MTT dye reduction assay using WST-8 (2-(2-methoxy-4-nitrophenyl)-3-(4-nitrophenyl)-5-(2,4-disulfophenyl)-2H-tetrazolium, monosodium salt) (Dojindo Molecular Technologies). Viable cell fractions were determined WST-8 values obtained from treated cells by using a microplate reader (Wallac 1420 ARVOsx).

**Cell fractionation.** Cells were collected and lysed in 0.25% Nonidet P-40 (PIERCE) for 10 min on ice. After centrifugation of the lysates at $12,000 \times g$ for 15 min at 4 °C, the pellet was pooled as the nuclear fraction and the supernatant as the cytosolic fraction. The fractionation efficacy was assessed by Western blotting for analyzing β-tubulin (cytoplasm) and lamin B (nucleus).

**PP2A immunoprecipitation phosphatase assay.** Assays for PP2A phosphatase activity bound to the ATM protein were performed by using a commercially available assay kit (Millipore). Briefly, FLAG-tagged ATM precipitates were washed with Ser/Thr assay buffer and incubated at 30 °C for 10 min after adding 750 μM phosphopeptide (K-R-pT-I-R-R). The precipitates were then centrifuged briefly, and Malachite Green Phosphate Detection Solution (100 μL, Millipore) was added to the supernatant (25 μL), and let color develop for 10–15 min at room temperature. The protein phosphatase activity was determined by reading the solution at 650 nm with the microplate reader (Wallac 1420 ARVOsx).

**Transfection of wild-type and mutant ATM DNA.** 293T cells were transfected with the wild-type (WT, *pcDNA3.1(+)Flag-His-ATMwt*, addgene), kinase-dead (KD, *pcDNA3.1(+)Flag-His-ATM kd*, addgene), or S1981A-mutated (S1981A, *hATMS1981A*, addgene) DNA. Transfection was performed using Lipofectamine 3000. Other ATM mutations at R533A, K2117A, or K2992A/S2996 were also studied for the effect on AKT binding to the ATM proteins in 293T cells (*pcDNA3.1(+)Flag-His-ATMR533A, -ATMK2117A, or-ATMK2992A/S2996A*).

**Spectroscopy.** Post-translational modifications (PTM) of ATM proteins were directly analyzed after in-gel digestion with trypsin, chymotrypsin, and esterase of the FLAG-tagged ATM proteins obtained from control, IR-treated (24 h), or hypoxic (5 h) 293T cells by liquid chromatography coupled with tandem mass spectroscopy (LC-MS/MS). Trypsin and chymotrypsin digestion was performed at 37 °C for 4 h and overnight, respectively, after washing with 25 mM ammonium bicarbonate by acetonitrile and then reducing with 10 mM dithiothreitol at 60 °C followed by alkylation with 50 mM iodoacetamide at room temperature. Quenching was performed with formic acid, and the supernatant was analyzed directly. LC-MS/MS was performed with a Waters NanoAcquity HPLC system interfaced to a Thermo Fisher Q Exactive. Peptides were loaded on a trapping column and eluted over a 75-μm analytical column at 350 nL/min. The mass spectrometer was operated in a data-dependent mode, with the Orbitrap operating

at 60,000 FWHM and 17,500 FWHM for MS and MS/MS, respectively. The 15 most abundant ions were selected for MS/MS. Data processing was performed using a local copy of Byonic (Protein Metrics), with variable modifications setting for acetyl (protein N-terminal, K), oxidation (M), deamidated (N, Q), phosphorylated (S, T, Y), methylated (K, R), dimethylated (K), and trimethylated (K). Scaffold results were exported as mzIdentML, and imported into Scaffold PTM in order to assign site localization probabilities using A-Score. A minimum localization filter of 505 was applied. Peak areas for detected modified peptides and their non-phosphorylated counterparts were calculated using Byologic (Protein Metrics).

**Statistics and reproducibility.** All data were calculated from at least three ($n = 3-6$) independent experiments and were represented as means ± s.e.m. All attempts at replication were successful with relatively small s.e.m. Sample size was chosen based on our previous experiments and publications (refs. [1] and [2], PMID: 19609276, PMID: 25880086). $P$-values were calculated with $t$ test for comparing two groups or Tukey–Kramer test for ≥3 groups using SPSS software (version 18, IBM). $P$-values < 0.05 were considered to be statistically significant, and the results were indicated as n.s. ($P > 0.05$) or providing the precise values. Western blot analysis was repeated at least three times, and representative data derived from consistent results within repeated experiments are shown in the main text or Supplementary Information.

**Reporting summary.** Further information on research design is available in the Nature Research Reporting Summary linked to this article.

## Data availability

All relevant data are included in the paper (Figs. 1–4) and its Supplementary information (Supplementary Figs. 1–8/Supplementary Table 1). Uncropped blots are shown in Supplementary Fig. 9. All source data underlying the graphs presented in the main figures are provided as Supplementary Data 1.

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

## Acknowledgements

This work was supported by research grants from Japanese Grant-in-Aid for Scientific Research (B) 24390421 to T.N., Japanese Grant-in-Aid for Challenging Exploratory Research 26670819 to T.N., Japanese Grant-in-Aid for Scientific Research (C) 16K11511 to Y.H., and Japanese Grant-in-Aid for Scientific Research (C) 19K10317 to I.K.

## Author contributions

T.N. conceived and supervised this study. Y.H. designed all experiments. Y.H. and I.K. cultured cells. Y.H. and I.K. performed western blotting. Y.H. performed immunoprecipitation assays and immunofluorescence microscopy. I.K. performed assays for cell viability. Y.H. and T.N. analyzed the data and prepared figures. T.N. wrote the paper with input from all authors.

## Competing interests

The authors declare no competing interests.
