## [Peer Review File · Communications Biology]

Reviewers' comments:

Reviewer #1 (Remarks to the Author):

The authors seek to understand the molecular mechanism underlying how cytoplasmic ATM functions as a protein platform to mediate Akt-dependent phosphorylation of GSK3 to control ER-stress induced cell death. The paper is clearly written, however, the following concerns should be addressed before its publication at Communications Biology.

1. Figure 1a, the authors should explain why pPERK, pIRE1a and pJNK signals peak at different time following CPT or IR stimulation. For example, pPERK peaks 6h post-CPT treatment while pPERK continues to rise 24-72 hours post-IR treatment?
2. Figure 1d, it is also important to include pAkt blot to exclude the possibility that GSK3i can feedback regulate Akt activity.
3. Figure 2b, it will be important to know whether GSK3i can restore Akti-induced degradation of NAC or Taxilin and induction of apoptosis. In addition, to exclude potential off-target effects of Akt inhibitor, it will be nice for the authors to examine whether siRNA-mediated depletion of Akt can achieve similar phenotypes of Akti.
4. Figure 3c, the authors should explain why after 24 hours of IR treatment, there is a sharp increase of Akt expression levels? Have the authors monitored Akt and pAkt in Figure 1a? Based on the results obtained in Fig. 2a, neither CPT, nor IR, Hypoxic treatment will affect AKT expression, which is different from being observed in Figure 3c.
5. Figure 3h vs Figure 1a, in Figure 1a, IR or CPT will lead to DNA damage and activation of ATM, while in Figure 3h, KU-60019 will lead to inactivation of ATM which is opposing to Figure 1a. The authors should explain why in both Figure 1a and Figure 3h, there is elevated JNK and induction of apoptosis, should the authors observe opposing phenotypes?

Reviewer #2 (Remarks to the Author):

This manuscript presents a series of data showing that PP2A on ataxia telangiectasia mutated (ATM) dephosphorylates AKT but not PI3K in response to CPT, IR, or etoposide treatment. This results in the activation of GSK3 β , followed by the induction of apoptosis-associated events, such as the down-regulation of α NAC and γ -taxilin, and the unfolded protein response (UPR). While the authors present extensive findings that show strong correlations of the above-mentioned events during CPT, IR, or etoposide treatment, these remain correlations and, unfortunately, do not provide mechanistic insights. Furthermore, the manuscript does not address how the cytosolic signaling events such as GSK3 β activation by PP2A intersect with the UPR signaling pathway that is thought to regulate endoplasmic reticulum functional homeostasis. In the current form, the work described here is not suitable for the general interest of readers of Nature Communication and would be more appropriate for a specialized journal.

Reviewer #3 (Remarks to the Author):

This work is focused in investigating the possible participation of ATM in ER stress-induced apoptosis. This study is well performed and several biochemical approaches were used to prove the hypothesis. Also, some of the observations are very interesting. However, the paper abuse of correlation without a clear mechanistic insights and lacks an extended discussion about their results. Also some of the conclusions are overestimated as the direct regulation of ATM, p-GSK3 β

and AKT in the ER stress –UPR-Cell dead response sometimes needs more controls. The data regarding the interaction between AKT/ATM in the cytosol could be improved with other techniques like proximity ligation assay to validate the observations and define the main location of this interaction occurred. If the authors are suitable to answer the following queries the paper would be recommended to be published in the journal.

Does the down regulation of NAC and γ TX with complementary approaches (ie siRNA/ShRNA) trigger ER stress and terminal UPR?

Are all the UPR branches turned on the experimental model? For example, is it possible to detect the generation of XBP1s or mRNA decay by RIDD mechanisms? Also, the phosphorylation of IRE1 is not convincing, please improve the data.

Why the authors didn't try ATM KO cells to determine if the effect on UPR and cell dead?

The link between ATM and AKT is not well documented. Is PIKK enzyme the only one responsible of the the pro apoptotic pathway?

Are the effectors downstream of ATM activated under DNA damage in the model? CHK2, P53 (and its pro-apoptotic targets)?

The activation of GSK3 β could lead the activation of others pro-apoptotic signals such as bax, tip60/p53 that could directly promote cell death. Is this mechanism involved in the study?

Minor comments

-Improve the cytometry graphs

-Specify the number of independent replicates in figure legends

-Correct some typograph error as "but not phsophorylated AKT and trace amounts of" page 6

The comments raised by the reviewers have been very helpful in revising our manuscript. We have attempted to address the questions according to the following:

Reviewer 1

1. Figure 1a, the authors should explain why pPERK, pIRE1a and pJNK signals peak at different time following CPT or IR stimulation. For example, pPERK peaks 6h post-CPT treatment while pPERK continues to rise 24-72 hours post-IR treatment?

Response. As the reviewer indicated, the kinetics of PERK, IRE1 α , and JNK activation after CPT treatment (peaks at ~6 h) was distinct from that observed after IR treatment (peaks at ~24-72 h) (Fig. 1a). The difference in kinetics may reflect the magnitude of ER stress impacts on the cell. Consistent with this idea, the annexin-positive cell ratios were inversely correlated with the times required for PERK, IRE1 α and JNK activation to reach their peaks in HeLa S3 cells, with the ratios of ~12% at 6 h and ~77% at 24 h after CPT treatment (p-PERK, p-IRE1 α and p-JNK peaks at ~6 h); and the ratios of ~21% at 24 h, ~40% at 48 h and ~61% at 72 h after IR treatment (peaks at ~24-72 h). Correlations between time courses of UPR-related protein activation and the cell's ultimate fate under ER stress have been suggested, with an emphasis on the critical timing of IRE1 activation and its termination in determining cell death fate (Science 318, 944-949, 2007 [ref. 7]). Consistent with this hypothesis, the timing of IRE1 α activation/termination as monitored with p-IRE1 α protein levels was correlated with the annexin-positive cell fractions in cells under ER stress (Fig. 1a, b).

In the revised version (page 3), we have noted the different kinetics of ER stress response protein activation between different types of inducers, and have discussed the correlation between the kinetics profile and the cell fate.

2. Figure 1d, it is also important to include pAkt blot to exclude the possibility that GSK3i can feedback regulate Akt activity.

Response. To investigate the feedback regulation of AKT by GSK3 β inhibition, we analyzed p-AKT expression levels in CPT-or IR-treated HeLa S3 cells in the presence or absence of a GSK3 β inhibitor (LiCl or CHIR). We found that the GSK3 β inhibition upregulates p-AKT in CPT- or IR-treated cells (Fig. 2f). These results suggest that feedback-regulated AKT activation contributes to the rescue of ER-stressed cells from apoptotic cell death.

In the revised manuscript, we have added AKT/p-AKT blots (Fig. 2f) from CPT- or IR-treated HeLa S3 cells following GSK3 inhibition and have discussed the results briefly as above (page 5).

3. Figure 2b, it will be important to know whether GSK3i can restore Akti-induced degradation of NAC or Taxilin and induction of apoptosis. In addition, to exclude potential off-target effects of Akt inhibitor, it will be nice for the authors to examine whether siRNA-mediated depletion of Akt can achieve similar phenotypes of Akti.

Response. To further evaluate the role of GSK3 β in the AKT inhibition-induced degradation of α NAC or γ TX and induction of apoptotic cell death, HeLa S3 cells were treated with perifosine in the presence or absence of CHIR, and then α NAC and γ TX protein expression levels and apoptotic cell ratios were monitored. We found that the GSK inhibitor restored α NAC and γ TX protein expression levels, and suppressed caspase-9 activation and apoptotic cell death of perifosine-treated cells (Fig. 2d, e), suggesting a pivotal role of GSK3 β in the AKT inhibition-induced apoptotic pathway.

To exclude potential off-target effects of the pharmacological AKT inhibitors, AKT was separately depleted by RNA interference in HeLa S3 cells. We found that the siRNA-mediated AKT depletion substantially suppresses GSK3 β phosphorylation, downregulates α NAC and γ TX, and induces CHOP and JNK activation (Fig. 2g). However, these changes were not observed in cells treated with a control siRNA or in cells that were mock treated. Furthermore, the AKT siRNA, but not mock treatment or control siRNA, significantly accelerated apoptotic cell death (Fig. 2h). Therefore, the siRNA-mediated AKT inhibition triggered ER stress responses similar to those caused by the pharmacological AKT inhibitor. These results support the notion that AKT acts as an upstream regulator of GSK3 β activation leading to the α NAC and γ TX depletion-mediated ER stress responses and cell death pathway.

In the revised version, we have provided these data (Fig. 2d, e, g, h) and explained the results (page 4–5).

4. Figure 3c, the authors should explain why after 24 hours of IR treatment, there is a sharp increase of Akt expression levels? Have the authors monitored Akt and pAkt in Figure 1a? Based on the results obtained in Fig. 2a, neither CPT, nor IR, Hypoxic treatment will affect AKT expression, which is different from being observed in Figure 3c.

Response. The cell lines used for these two figures are different; HeLa S3 cells were used for the Fig. 2a, while 293T cells were used for the original Fig. 3c (Fig. 3d in the revised version).

As the reviewer indicates, the kinetics of total AKT protein levels is different between HeLa S3 and 293T cells. To avoid the confusion, we have added a brief note of using different cell lines between these figures (Fig. 2a vs. Fig. 3d). The rephrased part has appeared on page 6 of the revised version as “Intriguingly, the kinetics of total AKT protein levels is distinct between HeLa S3 (Fig. 2a) and 293T cells (Fig. 3d), with sharp increases in the AKT levels being observed 5-24 h after ER stress induction in 293T cells, as opposed to HeLa S3 cells without such upregulation kinetics of total AKT, but rather the protein unchanged throughout the time courses. Nevertheless, p-AKT was similarly downregulated in both the cell lines that were treated with different types of ER stress inducers.” We used 293T cells for the data in Fig. 3d, e, f, g and h; Fig. 4a, b, c, d and f; and Extended Data Fig. 8. Otherwise we used HeLa S3 cells. We have indicated the cell lines used (HeLa S3 or 293T cells) within all the figure legends.

The observed difference in the AKT expression levels between these cell lines may be due to a possible difference in growth rates between these cell lines. In general, AKT activation promotes cell survival, proliferation and growth (Cell 169, 381-405, 2017 [ref. 9]). Therefore, we speculate that the excess of unphosphorylated over phosphorylated AKT in non-stressed HeLa S3 cells (Fig. 2a) implies cellular machinery intrinsically programmed against the cell’s propensity for high proliferation rates (J Exp Med 104, 427-434, 1956 [ref. 15]), opposing to 293T cells, in which phosphorylated AKT is predominant over unphosphorylated AKT (Fig. 3d). Upon ER stress, 293T cells may respond the stress by means of decreasing phosphorylated AKT and simultaneously by increasing unphosphorylated AKT in order to effectively suppress the the cell proliferation, while HeLa S3 cells can achieve this by downregulating phosphorylated AKT without the need for upregulating AKT. Although still conjectural, we have also added this notion in the revised manuscript (page 6-7) in response to the reviewer’s comment.

5. Figure 3h vs Figure 1a, in Figure 1a, IR or CPT will lead to DNA damage and activation of ATM, while in Figure 3h, KU-60019 will lead to inactivation of ATM which is opposing to Figure 1a. The authors should explain why in both Figure 1a and Figure 3h, there is elevated JNK and induction of apoptosis, should the authors observe opposing phenotypes?

Response. The kinetics of phosphorylated (S1981) ATM protein has been summarized in Extended Data Fig. 7 in the revised version. As expected, DNA damaging agents (CPT, IR and etoposide) all induced ATM phosphorylation at Ser1981 in HeLa S3 cells during their early time courses, whereas KU-60019 treatment was not associated with this hallmark of DNA damage-induced ATM activation (Fig. 3i and Extended Data Fig. 7), instead provoking caspase-dependent protein degradation after ER stress induction (Fig. 3i, k, and Extended Data Fig. 6). Therefore, the DNA-damaging agents and KU-60019 exerted opposing effects on the ATM activity. Nevertheless, both the treatment types similarly enhanced UPRs and JNK

phosphorylation, and comparably induced apoptotic cell death (Figs. 1a, b and 3i, j). These results suggest that ER stress responses leading to cell death can occur without activating ATM protein, at least not necessarily requiring the canonical phosphorylation at Ser1981 unlike in DNA damage responses. In addition, DNA damage-induced ATM activation and the subsequent recruitment of the activated protein to the damaged DNA sites occurs at early stages ($< \sim 1$ h) of DNA damage responses in the nucleus (Mol. Cell 66, 801-817, 2017 [ref. 21]), whereas the UPRs become evident in the cytoplasm at later ($> \sim 6$ h) stages after ER stress induction by DNA damaging agents or KU-60019 (Figs. 1a and 3i). Therefore, the early DNA repair events in the nucleus are seeking to rescue the cell from cell death by means of activating the nuclear ATM (Nat. Rev. Mol. Cell Biol. 14, 197-210, 2013 [ref. 13]); however, the UPR events occurring at later stages in the cytoplasm may use the cytoplasmic ATM as a cofactor or a platform, directing the cell to cell death processes when the earlier nuclear events cannot ameliorate the damaged states in the cell.

In the revised manuscript (page 8–9), we have added the above comments concerning the similar kinetics of ER stress response-related proteins in spite of different responses of ATM activation between the DNA damaging agents- and KU-60019-treated cells.

Reviewer 2

While the authors present extensive findings that show strong correlations of the above-mentioned events during CPT, IR, or etoposide treatment, these remain correlations and, unfortunately, do not provide mechanistic insights. Furthermore, the manuscript does not address how the cytosolic signaling events such as GSK3 β activation by PP2A intersect with the UPR signaling pathway that is thought to regulate endoplasmic reticulum functional homeostasis.

Response. We have attempted to address the comments raised by the Reviewer 2 by performing additional experiments that were also suggested by the other reviewers, according to the following:

(a) To investigate the feedback regulation of AKT by GSK3 β inhibition, we assessed p-AKT expression levels in CPT- or IR-treated HeLa S3 cells in the presence or absence of a GSK3 β inhibitor (LiCl or CHIR). We found that GSK3 β inhibition upregulates p-AKT in the CPT- or IR-treated cells (Fig. 2f). These data suggest that feedback-regulated AKT activation contributes to the rescue of ER-stressed cells from apoptotic cell death. (page 5, Reviewer 1/Comment 2)

(b) To further evaluate the role of GSK3 β in the AKT inhibition-induced degradation of α NAC or γ TX and induction of apoptotic cell death, HeLa S3 cells were treated with perifosine in the

presence or absence of CHIR, and then α NAC and γ TX protein expression levels and apoptotic cell ratios of these cells were monitored. We found that the GSK inhibitor restores α NAC and γ TX protein expression levels, and suppresses caspase-9 activation and apoptotic cell death of perifosine-treated cells (Fig. 2d, e). These data support a pivotal role of GSK3 β in the AKT inhibition-induced apoptotic pathway. (page 4-5, Reviewer 1/Comment 3)

(c) To exclude off-target effects of the pharmacological AKT inhibitors, AKT was separately depleted by RNA interference. We found that siRNA-mediated AKT depletion substantially suppresses the p-GSK3 β levels, downregulates α NAC/ γ TX, induces CHOP and JNK activation, and significantly accelerates apoptotic cell death (Fig. 2g, h). These results further support the notion that AKT acts as an upstream regulator of GSK3 β activation leading to the α NAC and γ TX depletion-mediated ER stress responses and cell death pathway. (page 5, Reviewer 1/Comment 3)

(d) Cell fractionation analysis and confocal microscopy indicated the subcellular compartmentalization of ATM and AKT proteins and the colocalization of these proteins (Fig. 3a, b). However, these data may be insufficient to define the subcellular interaction sites for these proteins. To this end, we performed proximity ligation assay (PLA). PLA indicated that these proteins mainly interact with each other at the perinuclear cytoplasm (Fig. 3c). (page 5-6, Reviewer 3/Comment 1)

(e) A complementary approach to α NAC/ γ TX downregulation via RNA interference could be supportive for the idea that downregulation of these proteins can trigger ER stress responses. We did not perform this in the present study; however, our siRNA data that were presented in previous studies (Cell Death Differ 16, 1505-1515, 2009 [ref. 1], Cell Death Dis 6, e1719, 2015 [ref. 2]) showed that siRNA-mediated α NAC or γ -taxilin depletion can also trigger UPRs. These data support the idea that the α NAC/ γ TX downregulation can trigger ER stress responses. We have referred these results in the revised manuscript. (page 2, Reviewer 3/Comment 2)

(f) Additional Western blot analysis shows upregulation of XBP1s in CPT- or IR-treated HeLa S3 cells (Fig. 1a). Thus, all the UPR branches are turned on in the present experimental model. (page 2, Reviewer 3/Comment 3)

(g) p-Chk2 (Thr68) and p-p53 (Ser15) were upregulated in CPT- or IR-treated HeLa S3 cells (Extended Data Fig. 2), confirming that these effectors downstream of ATM were activated under DNA damage. (page 2, Reviewer 3/Comment 5)

(h) GSK3 β -mediated Tip60 phosphorylation has been implicated in the induction of apoptosis through the Puma/Bax axis (Mol Cell 42, 584-596, 201 [ref. 4]), and Tip60-dependent p53 acetylation can induce apoptosis via increased mitochondrial membrane permeability (Mol

Cell 24, 827-839, 2006 [ref. 5]; Cell 149, 1536-1548, 2012 [ref. 6]). These findings imply that Tip60 and Bax are activated in the ER stress-induced apoptotic pathway. We therefore tested this possibility in HeLa S3 cells treated with CPT or IR, and found that CPT and IR both induce Tip60 phosphorylation and Bax upregulation (Fig. 1a). By contrast, p-Tip60 was not upregulated in hypoxic cells (Extended Data Fig. 2b). Thus, the present data do not fully support the involvement of Tip60 in the ER stress-induced apoptotic pathway. (page 2-3, Reviewer 3/Comment 6)

(i) PKC signalling could be another potential effector upstream of AKT in the ER stress-induced pro-apoptotic pathway, since some types of PKC family proteins can phosphorylate AKT and GSK3 β (JBC 288, 3918-3928, 2013 [ref. 10]). To test this possibility, we asked whether PKC inhibition could activate AKT/GSK3 β / α NAC/ γ TX-dependent apoptotic pathway. We found that PKC α and PKC δ expression levels are not changed in cells under ER stress caused by CPT, IR, or hypoxia (Extended Data Fig. 4a). Moreover, PKC inhibition by a pan-PKC inhibitor sotrastaurin did not downregulate p-AKT, p-GSK3 β , α NAC, or γ TX, and the inhibition did not accelerate apoptotic cell death (Extended Data Fig. 4b, c). These data indicate that the PKC signalling is unlikely to be involved in the ER stress-induced pro-apoptotic pathway. (page 4, Reviewer 3/Comment 4)

(j) Finally, we have clearly stated the results from siRNA-mediated ATM knockout (KO) experiments. We found that the ATM ablation causes downregulation of p-AKT, p-GSK3 β , α NAC and γ TX, and triggers UPRs such as CHOP and JNK activation, findings mimicking the changes with CPT, IR, hypoxic stress, or KU-60019 (Extended Data Fig. 5a vs. Figs. 1a and 3i, and data from Cell Death Differ 16, 1505-1515, 2009 [ref. 1] and Cell Death Dis 6, e1719, 2015 [ref. 2]). These data further define the link between ATM and AKT in ER stress responses. However, apoptosis induction by the siRNA-mediated ATM KO was less effective compared with the other ER stress inducers tested: ~ 33% for ATM KO vs. ~ 93% for CPT, ~61% for IR, and ~ 73% for KU-60019 in HeLa S3 cells (Extended Data Fig. 5b vs. Fig. 1b, Fig. 3j). These data imply differential cell death impacts between the pharmacological inhibition of ATM and the siRNA-mediated ATM protein ablation. (page 8, Reviewer 3/Comment 4)

Reviewer 3

1. The data regarding the interaction between AKT/ATM in the cytosol could be improved with other techniques like proximity ligation assay to validate the observations and define the main location of this interaction occurred.

Response. According to the suggestion, we performed proximity ligation assay (PLA) to define the main subcellular interaction sites of ATM and p-AKT. In the revised version, we have

added PLA data (Fig. 3c), discussed the results and rephrased related parts of the text (page 5-6) as follows:

Confocal microscopy of untreated HeLa S3 cells showed that ATM is uniformly distributed in both the cytoplasm and nucleus, except for the nucleoli (Fig. 3a). By contrast, p-AKT predominantly localizes to the cytoplasm. Thus, p-AKT colocalizes with ATM in the cytoplasm and to a lesser extent within the nucleus (Fig. 3a, upper panel). Western blot analysis of fractionated cell lysates provided data consistent with the findings by confocal microscopy (Fig. 3b). To further interrogate the main interaction site of ATM and p-AKT proteins, we performed proximity ligation assay (PLA). We found that PLA fluorescence puncta localize to both the cytoplasm and nucleus of HeLa S3 cells, with most PLA signals detected in the perinuclear region of the cytoplasm or in the peripheral part of the nucleoplasm (Fig. 3c). Considering the 3D confocal PLA imaging with z-stacking and maximum intensity projection (MIP), where the perinuclear cytoplasm and the nuclear periphery may overlap on stacked images, these results suggest that the interactions between ATM and p-AKT proteins occur predominantly in the perinuclear region of the cytoplasm.

A brief methodology for the PLA has been provided in page 12.

2. Does the down regulation of NAC and γ TX with complementary approaches (ie siRNA/ShRNA) trigger ER stress and terminal UPR?

Response. In the present study, we did not perform siRNA interference of α NAC or γ TX. However, our previous studies using the same experiment models as in the present study showed that the siRNA-mediated α NAC or γ TX depletion can trigger UPRs and accelerate cell death (Cell Death Differ 16, 1505-1515, 2009 [ref. 1]; Cell Death Dis 6, e1719, 2015 [ref. 2]). We have referred to these results in the revised version (page 2).

3. Are all the UPR branches turned on the experimental model? For example, is it possible to detect the generation of XBP1s or mRNA decay by RIDD mechanisms? Also, the phosphorylation of IRE1 is not convincing, please improve the data.

Response. We have added Western blots for XBP1s in CPT- or IR-treated cells (Fig. 1a) and a brief comment on the results (page 2) in the revised manuscript. In addition, the Western blots for p-IRE1 α have been improved by using an anti-p-IRE1 α antibody obtained from a company different from that in the original version (Fig. 1a, d).

4. Why the authors didn't try ATM KO cells to determine if the effect on UPR and cell dead? The link between ATM and AKT is not well documented. Is PIKK enzyme the only one responsible of the the pro apoptotic pathway?

Response. In the original version, we had assessed the effects of siRNA-mediated ATM knockout (KO) on UPRs and cell death, and found that the ATM ablation causes downregulation of p-AKT, p-GSK3 β , α NAC and γ TX, and triggers UPRs such as CHOP and JNK activation, findings mimicking the changes with CPT, IR, hypoxic stress, or KU-60019 (Extended Data Fig. 5a vs. Figs. 1a and 3i, [all in the revised version] and data from Cell Death Differ 16, 1505-1515, 2009 [ref. 1] and Cell Death Dis 6, e1719, 2015 [ref. 2]). These data further define the link between ATM and AKT in ER stress responses. However, apoptosis induction by the siRNA-mediated ATM KO was less effective compared with the other ER stress inducers tested: ~ 33% for ATM KO vs. ~ 93% for CPT, ~61% for IR, and ~ 73% for KU-60019 in HeLa S3 cells (Extended Data Fig. 5b vs. Fig. 1b and Fig. 3j [all in the revised version]). These data imply differential cell death impacts between the pharmacological inhibition of ATM and the siRNA-mediated ATM protein ablation.

The above explanation was almost totally lacked in the original manuscript. In the revised version, we have explained these results on page 8.

PKC signalling could be another potential effector upstream of the AKT protein in the ER stress-induced pro-apoptotic pathway, since some types of PKC family proteins can phosphorylate AKT and GSK3 β (JBC 288, 3918-3928, 2013 [ref. 10]). To test this possibility, we asked whether PKC inhibition could activate AKT/GSK3 β / α NAC/ γ TX-dependent apoptotic pathway. We found that PKC α and PKC δ expression levels are not changed in cells under ER stress caused by CPT, IR, or hypoxia (Extended Data Fig. 4a). Moreover, PKC inhibition by sotrastaurin did not downregulate p-AKT, p-GSK3 β , α NAC, or γ TX, and the inhibition did not accelerate apoptotic cell death (Extended Data Fig. 4b, c). These data indicate that the PKC signalling is unlikely to be involved in the ER stress-induced pro-apoptotic pathway.

We have explained the data in page 4 of the revised version.

5. Are the effectors downstream of ATM activated under DNA damage in the model? CHK2, P53 (and its pro-apoptotic targets)?

Response. As expected, p-Chk2 (Thr68) and p-p53 (Ser15) were upregulated in CPT- or IR-treated HeLa S3 cells, confirming that these effectors downstream of ATM were activated under DNA damage. (Extended Data Fig. 2a).

We have provided Western blots for Chk2 and p53 in Extended Data Fig. 2a, and have explained the results in page 2.

6. The activation of GSK3 β could lead the activation of others pro-apoptotic signals such as bax, tip60/p53 that could directly promote cell death. Is this mechanism involved in the study?

Response. GSK3 β -mediated Tip60 phosphorylation has been implicated in the induction of apoptosis through the Puma/Bax axis (Mol Cell 42, 584-596, 201 [ref. 4]), and Tip60-dependent p53 acetylation can induce apoptosis via increased mitochondrial membrane permeability (Mol Cell 24, 827-839, 2006 [ref. 5]; Cell 149, 1536-1548, 2012 [ref. 6]). These findings suggest that Tip60 and Bax are activated in the ER stress-induced apoptotic pathway. We therefore tested this possibility in HeLa S3 cells treated with CPT or IR, and found that CPT and IR both induce Tip60 phosphorylation (Ser86) and Bax upregulation (Fig. 1a). By contrast, p-Tip60 was not upregulated in hypoxic cells (Extended Data Fig. 2b). Thus, the present data do not fully support the involvement of Tip60 in the ER stress-induced apoptotic pathway.

We have provided the Tip60 data in the Fig. 1a and Extended Data Fig. 2b and the Bax data in Fig. 1a, and have explained the results in page 2-3 of the revised manuscript.

Minor comments

7. Improve the cytometry graphs

Response. We have improved the cytometry graphs (histograms) by clearly defining the X- and Y-axes, providing numerical data (% of annexin V-positive cell fractions) just below the corresponding horizontal bars in the graphs, and also by color-coding the graphs in accordance with the corresponding bar graphs (Fig. 1b, Fig. 2c, e, h, Fig. 4e, and Extended Data Fig. 5b).

8. Specify the number of independent replicates in figure legends

Response. We have added the number of independent replicates in the legends for Fig. 1b, e-h, Fig. 2c, e, h, Fig. 3j, Fig. 4d, e, Extended Data Fig. 4c, and Extended Data Fig. 5b.

9. Correct some typograph error as “but not phsophorylated AKT and trace amounts of” page 6

Response. We have corrected typographical errors (‘phosphorylated AKT’ and ‘phosphorylation at Ser473’ on page 10 and ‘phosphorylation’ and ‘methylation’ in Extended Data Table 1).

Finally, the discussion regarding the AKT signalling pathways in the cytosol (page 10) is speculative without definite evidence provided in the present study. Thus, we have deleted this part from the revised version.

The detailed review of this manuscript is appreciated and we have attempted to answer each of the questions raised.

Thank you for your consideration of the revised version.

REVIEWERS' COMMENTS:

Reviewer #1 (Remarks to the Author):

The authors have addressed most of the raised concerns during this round of revision.

Reviewer #3 (Remarks to the Author):

I appreciate the work done by the research group. All the new experiments really improve the quality of the work. However, I still have some concerns about the main conclusion expressed:

"The ATM Protein is a Cytoplasmic Platform for Endoplasmic Reticulum Stress-Induced Cell Death"

1. If this is true the ATM knockdown or the inhibition of its cytoplasmic localization in the context of the ER stress induced by several known ER stressors (as thapsigargin, tunicamycin, expression of protein prone to misfold, etc) has to show a significant effect in the ER Stress-Induced Cell Death. However, the researchers, mainly focused on the DNA damage as the main stressors stimuli. So, the ATM role is particular of the DNA damage and the ER-stress cell death phenotype is secondary.

2. Also, the inhibition of the UPR sensors under IR/CPT and its consequences in the cell death were not investigated by the group. So, I suggest to check the title and the results to clearly express the main conclusion in the title.

3. CHOP expression and p-JNK are not only output of the Endoplasmic Reticulum Stress-Induced Cell Death

4. The ER stress induced by the exposure of the DNA damage agents and the engage of the Endoplasmic Reticulum Stress-Induced Cell Death is the main mechanism of the apoptosis? What about with the DNA Damage cell death pathway?

Also, the use of RNAi is known as Knockdown the use of knockout word it is incorrect

If the authors clarify these concerns the work will be suitable for publication.

Reviewer 3

Comments 1 and 2. *I still have some concerns about the main conclusion expressed: “The ATM Protein is a Cytoplasmic Platform for Endoplasmic Reticulum Stress-Induced Cell Death”. (1) If this is true the ATM knockdown or the inhibition of its cytoplasmic localization in the context of the ER stress induced by several known ER stressors (as thapsigargin, tunicamycin, expression of protein prone to misfold, etc) has to show a significant effect in the ER Stress-Induced Cell Death. However, the researchers, mainly focused on the DNA damage as the main stressors stimuli. So, the ATM role is particular of the DNA damage and the ER-stress cell death phenotype is secondary. (2) Also, the inhibition of the UPR sensors under IRE1/CPT and its consequences in the cell death were not investigated by the group. So, I suggest to check the title and the results to clearly express the main conclusion in the title.*

Response. According to the suggestion by the Reviewer 3 and the Editor, we have rewritten the title as “ATM-associated signalling triggers the unfolded protein response and cell death in response to stress”.

Comment 3. *CHOP expression and p-JNK are not only output of the Endoplasmic Reticulum Stress-Induced Cell Death.*

Response. As mentioned above, we have rewritten the title so as not to mislead the readers into thinking that we consider CHOP expression and p-JNK to be only output of the ER stress-induced cell death. In addition, we have rephrased the related sentences (on page 4, in the second paragraph) to “To monitor apoptotic cell death, we chose CHOP, p-JNK, Bax and cleaved caspase-9 (c-casp-9) among many proteins involved in the ER stress-induced cell death processes”.

Comment 4. *Also, the use of RNAi is known as Knockdown the use of knockout word it is incorrect*

Response. We have replaced the term “knockout” with “ablation” for ATM RNAi (page 10).